# Unveiling the Role of SlRNC1 in Chloroplast Development and Global Gene Regulation in Tomato Plants

**DOI:** 10.3390/ijms25136898

**Published:** 2024-06-24

**Authors:** Yuxin Nie, Yuhong Zhang, Luyou Wang, Jian Wu

**Affiliations:** State Key Laboratory for Managing Biotic and Chemical Threats to the Quality and Safety of Agroproducts, Key Laboratory of Biotechnology in Plant Protection of MARA and Zhejiang Province, Institute of Plant Virology, Ningbo University, Ningbo 315211, China; 15169312489@163.com (Y.N.); yuhongzhang0105@139.com (Y.Z.); wly13858076596@163.com (L.W.)

**Keywords:** RNC1 protein, chloroplasts, tomato plants, RNA-seq

## Abstract

*RNC1*, a plant-specific gene, is known for its involvement in splicing group II introns within maize chloroplast. However, its role in chloroplast development and global gene expression remains poorly understood. This study aimed to investigate the role of *RNC1* in chloroplast development and identify the genes that mediate its function in the development of entire tomato plants. Consistent with findings in maize, *RNC1* silencing induced dwarfism and leaf whitening in tomato plants. Subcellular localization analysis revealed that the RNC1 protein is localized to both the nucleus and cytoplasm, including the stress granule and chloroplasts. Electron microscopic examination of tomato leaf transverse sections exposed significant disruptions in the spatial arrangement of the thylakoid network upon *RNC1* silencing, crucial for efficient light energy capture and conversion into chemical energy. Transcriptome analysis suggested that *RNC1* silencing potentially impacts tomato plant development through genes associated with all three categories (biological processes, cellular components, and molecular functions). Overall, our findings contribute to a better understanding of the critical role of *RNC1* in chloroplast development and its significance in plant physiology.

## 1. Introduction

Plant organelles have a fascinating origin, stemming from two distinct endosymbiotic events [1,2]. Over the course of evolution, these organelles, namely mitochondria and plastids, have engaged in a complex dance of genetic material exchange with the nucleus [3]. This process, known as compartmental DNA transfer, has led to the gradual loss of much of their original genomes. As a result of this genetic exchange, chloroplasts have emerged as semi-autonomous entities within plant cells, harboring their own unique genetic material and gene expression systems. Their significance cannot be overstated, as they play a central role in an array of biochemical processes critical to plant survival. Chloroplasts are veritable powerhouses within the cell, orchestrating the synthesis of essential compounds such as amino acids, pigments, lipids, and plant hormones [4,5]. Beyond their biochemical prowess, chloroplasts also serve as key players in how plants perceive and respond to their environment. From sensing gravity to regulating stomatal activity and mounting defenses against pathogens [6,7,8,9], chloroplasts are integral to a plant’s ability to adapt and thrive in its surroundings. In essence, they are not just cellular components but dynamic hubs that bridge the gap between a plant and its ever-changing environment.

Efficient precursor processing is crucial for maintaining chloroplast translation, as multiple ribonucleases are involved in rRNA processing. Initially, RNA splicing of introns was facilitated by intron-encoded proteins (IEPs). However, in terrestrial plants, nearly all introns shed their IEPs, leaving only two mature enzymes encoded in the organelle genome, alongside a suite of nuclear-encoded splicing factors [10]. Predominantly, the introns found in the organelles of terrestrial plants are of the group II variety [11], which exhibit a catalytic mechanism akin to nuclear spliceosomes. Nevertheless, plant group II introns lose their inherent ability to self-splice and instead necessitate nuclear-coded proteins as cofactors. Given that the majority of plastid proteins are governed by nuclear genes, the nucleus holds dominion over most facets of chloroplast genome expression and metabolism, particularly RNA maturation, including RNA splicing, which heavily relies on nuclear-encoded splicing factors. This communication from the nucleus to the chloroplast is often referred to as a positive signal. Interestingly, the developmental state of chloroplasts can reciprocally influence the expression of nuclear genes. Across various stages of chloroplast growth and development, the nucleus dispatches an array of signals to the chloroplast, thus orchestrating the expression of chloroplast genes and regulating chloroplast development [12]. These signals transmitted from chloroplasts or plastids to the nucleus are labeled as plastid reverse signaling [13,14].

RNA maturation in chloroplasts is of paramount importance due to their indispensable roles in photosynthesis, retrograde signaling, and plant development [15,16]. Nucleus-encoded proteins, including the CAF2/CRS2 complex, CAF1/CRS2 complex, aPPR4, DUF860, and CRS1, have been identified as essential for group II intron splicing in maize (*Zea mays*) chloroplasts [17,18]. Ribonuclease III (RNase III) enzymes are known to exert significant influence over the processing and metabolism of transcripts transcribed within the nucleus. Their functions encompass pivotal aspects such as pre-mRNA splicing, mRNA localization, stability, and translation [19]. In addition to its role in the maturation of transcripts from the nucleus, RNC1, a plant-specific protein featuring two ribonuclease III (RNase III) domains, has been found to be involved in the splicing of transcripts originating from chloroplasts [17]. However, its role in the development of the chloroplast and its effects on global gene expression is unknown. This study aimed to explore the role of the RNC1 protein in the chloroplast development in tomato plants, and its role in regulating global gene expression.

## 2. Results

### 2.1. Identification of RNC1 in Tomato and Sequence Alignment RNC1 Proteins across Four Plants

Using the protein sequences of ZmRNC1 (maize), AtRNC1 (*Arabidopsis thaliana*), and OsRNC1 (rice) as queries, NCBI TBLASTN was employed to identify SlRNC1 (tomato). Protein sequences of ZmRNC1, AtRNC1, and OsRNC1 were obtained from a previous study [17]. Subsequently, the protein sequences of ZmRNC1, AtRNC1, OsRNC1, and SlRNC1 were aligned using DNAMAN. The multiple sequence alignment uncovered remarkably high similarity among these four proteins. The *SlRNC1* gene was obtained through BLAST analysis on NCBI, utilizing ZmRNC1, AtRNC1, and OsRNC1 RNC1 genes as queries. Notably, significant differences were observed in the N-terminal sequence, while the C-terminal sequence exhibited relatively high homology (Figure 1A). Furthermore, the structure of RNC1 in the four species was predicted using AlphaFold 2, successfully identifying two RNase III domains across all four plants, denoted by two black dashed-line circles. The structure of SlRNC1 is depicted (Figure 1B). As evidenced by the structural model, the unconserved N-terminal sequence exhibits a lack of well-defined structure, hinting at a potential unknown function, perhaps one that is species-specific.

### 2.2. VIGS-Induced SlRNC1 Silencing Results in Dwarf Tomato Plants with Yellowish Leaves

VIGS-mediated gene silencing was utilized to suppress the expression of *SlRNC1* to explore its role in the development of tomato plants. Two groups of tomato plants were included in this study: the tobacco rattle virus (TRV):00 (mock) group, comprising plants agroinfiltrated with pTRV2 (empty vector) along with pTRV1, and the TRV:*SlRNC1* group, consisting of plants agroinfiltrated with pTRV2-SlRNC1 along with pTRV1. Three biological replicates were conducted, each consisting of one plant. To evaluate the silencing of the *RNC1* gene, we analyzed total RNA extracted from systemic leaf samples of the agroinfiltrated plants at 14 dpi using RT-qPCR. Compared to the TRV:00 group, the mRNA levels of SlRNC1 decreased by 45% in TRV:*SlRNC1* tomato plants, indicating the successful silencing of this gene (Figure 2A). Phenotypes were recorded at 24 days post-agroinfiltration (dpi). Compared to plants in the TRV:00 group, those in the TRV:*SlRNC1* group exhibited a dwarf phenotype with small yellow leaves and much shorter internodes (Figure 2A). In addition, stems of some plants were also damaged at a very early stage. These results illustrate that silencing of *SlRNC1* resulted in dwarfing of tomatoes, yellowing of leaves, and damage to stems, all indicative of plant growth retardation. Our findings align with previous observations showing that the loss of function of RNC1 in maize resulted in pigment-deficient and dwarfing phenotypes [17].

### 2.3. Subcellular Localization of SlRNC1

To determine the cellular localization of the RNC1 protein, *SlRNC1* ORF was fused with GFP tags using the specialized localization vector p186-GFP, enabling the visualization of the protein within living cells. Utilizing the *A. tumefacient* mediated transient expression system, the resulting construct, p186-GFP-SlRNC1, was introduced into *N. benthamiana*, a model organism renowned for its experimental utility. To ensure the reliability of the findings, three biological replicates were conducted, each comprising three individual *N. benthamiana* plants. Following a brief incubation period of two days to allow for sufficient protein expression, the cellular localization of SlRNC1 was analyzed using a Leica confocal fluorescence microscope. The results revealed that p186-GFP-SlRNC1 exhibited localization in both the cytoplasm and nucleus. Its localization in the stress granule and chloroplasts was also observed (Figure 3). Our findings align with previous observations indicating that RNC1 can be found in the chloroplast stroma with CAF1 and CAF2 [17]. The observation of SlRNC1’s presence in additional cellular compartments suggests its involvement in a broader array of biological processes, expanding our understanding of its functional repertoire. However, further validation of the subcellular localization of RNC1 is warranted and can be achieved through alternative approaches, such as immunohistochemistry using antibodies specifically targeting the RNC1 protein.

### 2.4. Silencing of SlRNC1 Caused Ultrastructural Changes in Chloroplasts

To explore the involvement of the *SlRNC1* gene in chloroplast development, transmission electron microscopy (TEM) was employed to examine the ultrastructure of chloroplasts in leaves of plants inoculated with TRV:00 and TRV:*SlRNC1* at 24 days post-inoculation (dpi). The findings revealed that chloroplasts in plants from the TRV:00 group displayed an ellipsoidal shape with intact cell structure, devoid of plasmolysis indications, and exhibited a well-defined thylakoid structure composed of both stroma thylakoids and grana thylakoids. Furthermore, a small quantity of intracellular starch grains and a consistent arrangement of grana lamellae were observed (Figure 4A). In contrast, leaves of plants subjected to TRV:*SlRNC1* inoculation displayed a pronounced degree of chloroplast damage. Chloroplasts in these plants appeared smaller compared to those in TRV:00 plants and lacked the regular elliptical shape characteristic of normal chloroplasts. Furthermore, chloroplasts in TRV:*SlRNC1* plants exhibited disintegration, with damaged and blurred membrane structures. Notably, typical thylakoid membranes and stacking structures of grana were absent, and the layered structure appeared disordered, with a significant reduction in particles and expansion of the layered structure (Figure 4B). These findings suggest that silencing of the *SlRNC1* gene likely disrupts normal chloroplast development, resulting in chloroplast abnormalities and significant structural impairments. The observed chloroplast abnormalities and notable structural impairments are likely primarily attributed to inhibited chloroplast RNA splicing. However, it is important to acknowledge the potential involvement of other factors, such as alterations in the gene expression network, which may also contribute to these effects.

### 2.5. Analysis of Differentially Expressed Genes Caused by SlRNC1 Silencing

The raw data extracted from three samples in each of the TRV:00 and TRV:*SlRNC1* groups underwent a comprehensive bioinformatics analysis. A global gene expression profiling was meticulously conducted on data derived from leaves of tomato plants within the TRV:*SlRNC1* and TRV:00 groups (Figure 2B). This analytical approach aimed to unveil the intricate impact of *SlRNC1* silencing on the overall gene expression landscape. Remarkably, a high degree of consistency was observed across the data obtained from the three biological replicates of both the TRV:*SlRNC1* and TRV:00 groups. In total, the RNA-seq analysis generated an impressive 40.48 gigabases (Gb) of data, comprising a staggering 269,829,174 clean reads (Table 1). Notably, the quality metrics of the dataset were exemplary, with Q20 and Q30 ratios exceeding 98% and 95%, respectively, reaffirming the reliability and suitability of the dataset for subsequent in-depth analysis (Table 1). Delving deeper into the analysis, our findings revealed significant alterations in gene expression profiles between TRV:*SlRNC1* and TRV:00 leaves. Specifically, a comprehensive assessment identified 387 genes that exhibited downregulation and 730 genes that displayed upregulation in TRV:*SlRNC1* versus TRV:00 leaves (with a fold change [FC] > 2 and *p* < 0.05; Figure 5). This compelling evidence underscores the pivotal role of SlRNC1 in orchestrating molecular changes within the tomato plant, shedding light on its intricate regulatory mechanisms.

### 2.6. Gene Ontology (GO) and KEGG Analysis of DEGs

Gene ontology (GO) analysis is a powerful bioinformatics approach used to annotate and analyze the functional characteristics of genes and their products. According to the GO database, the biological functions of the differentially expressed genes (DEGs) were classified into three categories: molecular function (MF), cellular component (CC), and biological process (BP). In this study, the top 10 terms for each category are presented (Figure 6A), with blue, green, and orange representing MF, CC, and BP, respectively.

DEGs were further divided into upregulated and downregulated genes. The upregulated genes were primarily associated with the amide biosynthetic process, cellular amide metabolic process, peptide biosynthetic process, peptide metabolic process, and translation within the BP category, cytoplasm, cytoplasmic part, ribonucleoprotein complex, non-membrane-bounded organelle, intracellular non-membrane-bounded organelle, and ribosome within the CC category, and structural constituent of ribosome and structural molecule activity within the MP category (Figure 6B). In contrast, the downregulated genes were primarily associated with transcription regulator activity and DNA-binding transcription factor activity within the MF category (Figure 6C). Our data suggests that the silencing of RNC1 potentially regulates the expression of downstream genes, primarily by downregulating transcription factors and transcription regulators.

The Kyoto Encyclopedia of Genes and Genomes (KEGG) is a comprehensive database and resource for understanding the molecular functions and biological pathways of genes and gene products. In this study, KEGG analysis was also employed to identify enriched biological pathways compared to the whole transcriptome background. The genes upregulated in TRV:*SlRNC1* plants were particularly enriched in pathways such as ribosome biogenesis in eukaryotes, carbon fixation in photosynthetic organisms, and others (Figure 7A). Conversely, some downregulated genes were identified as key genes associated with plant hormone signal transduction and flavonoid biosynthesis (Figure 7B). In summary, SlRNC1 plays a crucial role in plant development by influencing the expression of nuclear genes related to ribosome biogenesis, photosynthesis, and chloroplast function.

### 2.7. qRT-PCR Confirmation of DEGs

To ensure the reliability of the DEGs identified through RNA-seq analysis, a rigorous validation process was undertaken. A selection comprising three upregulated genes and three downregulated genes, as determined by RNA-seq, underwent further scrutiny via qRT-PCR. Notably, the RNA samples utilized for qRT-PCR were identical to those employed in the initial RNA-seq analysis, ensuring a direct comparison between the two methodologies. Upon conducting qRT-PCR analysis, a striking concordance emerged between the findings of the two techniques. Specifically, the three genes identified as upregulated by RNA-seq exhibited consistent upregulation in the qRT-PCR results when compared to the control group (Figure 8A). Similarly, the three genes identified as downregulated via RNA-seq displayed a corresponding downregulation in the qRT-PCR analysis relative to the control group (Figure 8B). These congruent expression patterns observed across both RNA-seq and qRT-PCR platforms serve as compelling evidence of the robustness and accuracy of the RNA-seq data. The validation process not only reinforces the credibility of the DEGs identified but also underscores the reliability of the RNA-seq methodology employed in this study.

## 3. Discussion

The present study provides a comprehensive insight into the role of SlRNC1 in tomato plant development, particularly focusing on chloroplast function and gene expression regulation. Through a combination of molecular biology techniques and bioinformatics analyses, we have elucidated key aspects of SlRNC1-mediated regulation and its implications for plant growth and development. It is plausible that the disruption of chloroplast function induced by *SlRNC1* silencing might trigger widespread effects on nuclear gene transcription through plastid reverse signaling pathways [20,21,22], thereby impacting the growth of entire tomato plants.

Our findings regarding the identification and sequence alignment of RNC1 proteins across different plant species shed light on the evolutionary conservation and structural characteristics of this essential RNA-binding protein. The remarkable similarity observed among RNC1 proteins from maize, *Arabidopsis thaliana*, rice, and tomato underscores the conserved nature of this protein family [17]. Furthermore, the structural analysis using AlphaFold 2 revealed the presence of two RNase III domains in SlRNC1, consistent with its role in RNA processing and splicing. However, it is well established that RNC1 in maize lacks catalytic activity [17,23]. Hence, there is a need for a meticulous examination of the precise function of the RNase III domains in *RNC1* genes. Silencing of *SlRNC1* using VIGS resulted in notable phenotypic changes in tomato plants, including dwarfism, yellowing of leaves, and shortened internodes. These phenotypic alterations are indicative of disrupted plant growth and development, highlighting the pivotal role of SlRNC1 in maintaining normal physiological processes. Our observations are consistent with previous studies on maize, highlighting the conserved function of RNC1 across monocots and dicots [17]. This aligns with the conserved sequence and structure of RNC1s observed in different plant species. However, the non-conserved C-terminal regions of RNC1 proteins across plants may confer species-specific functions to this protein. Future studies could delve into analyzing the functional significance of this region.

RNC1, a member of the RNase III family encoded by nuclear genes, functions as an RNA-binding protein involved in the splicing of chloroplast RNAs [17]. Research indicates that the development of chloroplasts requires the coordinated regulation of nuclear genes, which must be co-expressed with chloroplast genes [24,25]. It has been well established that RNC1 plays a critical role in the splicing of chloroplast RNAs, while the effects of *RNC1* silencing on the development of chloroplasts have not been analyzed [17]. Ultrastructural analysis of chloroplasts in *SlRNC1*-silenced plants revealed significant abnormalities, including reduced size, disintegration of thylakoid membranes, and disruption of grana stacking. These structural impairments indicate compromised chloroplast development and functionality, which likely contributes to the observed phenotypic defects in SlRNC1-silenced plants. Our findings underscore the essential role of SlRNC1 in maintaining chloroplast integrity and function.

RNC1 is reported to be localized to the chloroplasts of maize. In this study, subcellular localization studies revealed the presence of SlRNC1 in both the cytoplasm and nucleus, with additional localization observed in chloroplasts. Localization determines function [26,27]. This dual subcellular localization suggests that, besides its involvement in the splicing of chloroplast RNAs, SlRNC1 may also participate in diverse cellular processes. Transcriptomic analysis using RNA-seq identified a set of DEGs in *SlRNC1*-silenced plants compared to control plants. GO and KEGG analyses revealed enrichment of pathways associated with ribosome biogenesis [28], carbon fixation [29], and plant hormone signaling in response to SlRNC1 silencing. Combining the results of subcellular localization analysis with RNA-seq data, our findings suggest that the SlRNC1 protein may directly interact with other cellular factors in the cytoplasm and nucleus to regulate the expression of other genes. Alternatively, SlRNC1 may primarily affect chloroplast development, potentially triggering plastid retrograde signaling to impact nuclear gene expression. It is worth noting that the downregulated genes are mainly transcription regulators and transcription factors [30] (Figure 6B), which may explain the severe phenotype of tomato plants with *RNC1* silencing. Stress granules are membraneless organelles found in the cytoplasm of cells. They are dynamic, temporary structures that form in response to various stresses experienced by the cell, such as heat shock, oxidative stress, viral infection, or nutrient deprivation [31,32]. The localization of RNC1 to stress granules suggests its involvement in stress responses and possibly regulatory mechanisms during tomato growth and development [33,34,35]. However, our conclusions warrant further validation through additional approaches, such as overexpression and gene knockout studies.

## 4. Materials and Methods

### 4.1. Plant Materials and Growth Conditions

*Solanum lycopersicum* (Tomato) plants (Cultivar: Rutgers) and *Nicotiana benthamiana Domin* (*N. benthamiana*) plants were cultivated within the controlled environment of a growth chamber at Ningbo University. Tomato seeds were kindly provided by Dr. Zhixiang Zhang at Chinese Academy of Agricultural Sciences. The chamber is equipped with supplementary lighting (5000 lx Lumilux Cool White lm) to maintain a temperature of 23 °C, a photoperiod of 14 h of light followed by 10 h of darkness, and a relative humidity of 60% [36]. Each experimental group comprised three biological replicates, with one plant per replicate. All experiments utilized plants with uniform growth conditions.

### 4.2. Agrobacterium Transformation-Mediated Transient Gene Expression

To induce transient expression in *N. benthamiana* plants using Agrobacterium tumefaciens (*A. tumefaciens*), transient expression vectors were introduced into the GV3101 strain via electroporation [9]. Following this, cultures of *A. tumefaciens* were suspended in an infiltration solution and applied to the underside of leaves. Leaf samples were then collected for assays 48 h after infiltration. The transfected agrobacteria were cultured overnight in LB medium containing 25 μg/mL kanamycin and 100 μg/mL rifampicin at 28 °C. Upon harvesting the next day, the agrobacteria were resuspended in an MMA solution (10 mM MgCl_2_, 100 μM acetosyringone, 10 mM MES pH 5.6) for subsequent experiments. All agroinfiltrations were conducted using an *A. tumefaciens* concentration of 0.5 OD.

### 4.3. RNA Extraction, RT-PCR, and RT-qPCR

The total RNA extraction was conducted using Trizol reagent following established protocols [36,37,38,39,40,41]. Subsequently, the extracted RNA underwent reverse transcription (RT) using the First Strand cDNA Synthesis Kit Rever Tra (TOYOBO, Osaka City, Japan). RT was performed at 37 °C for 15 min, followed by 85 °C for 5 s to inactivate the enzyme. Amplification of the genomes was achieved using KOD FX Neo high-fidelity DNA polymerase (TOYOBO, Japan) with specific primers. For quantitative PCR (qPCR), AceQ qPCR SYBR Green PCR Master Mix kit (Vazyme, Nanjing, China) was utilized, with the specificity of each reaction confirmed through melting curve analysis [42]. RT-qPCR was carried out on a LightCycler480 instrument (Roche, Basel, Switzerland) with 384-well PCR plates. The reaction mixtures comprised 18 μL of ChamQ Universal SYBR qPCR Master Mix (Vazyme, Nanjing, China), 0.5 μL of each primer (10 μM), 6 μL of cDNA template, and 11 μL of RNase-free water. Three biological replicates were included for each experiment, and three technical replicates were set for each biological replicate. The thermal cycling conditions for RT-qPCR were 95 °C for 30 s, followed by 40 cycles of 15 s at 95 °C, 14 s at 55 °C, and 28 s at 72 °C. Melting curves were prepared with the following thermal conditions: 95 °C for 10 s, 60 °C for 30 s, and 95 °C for 15 s. Relative gene expression was analyzed using the 2^−∆∆CT^ method [42]. All primers utilized in this study are listed in Appendix A.

### 4.4. Plasmid Construction and Inoculation

The pTRV vectors were utilized for virus-induced gene silencing (VIGS) [43]. Fragments of 300 bp from *RNC1* (Tomato) and *PDS* genes were designed using the VIGS tool (http://solgenomics.net/tools/vigs 12 January 2024) to avoid off-target silencing. These fragments were then amplified and inserted into the pTRV2 vector following the manufacturer’s instructions. Cotyledons of tomato seedlings at the two- to four-leaf stage were inoculated with TRV:*SlRNC1* using *A. tumefaciens*-mediated inoculation, while an empty pTRV vector (TRV:00) served as a mock control [9]. For subcellular localization analysis, primers were designed to amplify the full-length SlRNC1 cDNA without a stop codon, which was subsequently incorporated into the green fluorescent protein (GFP) expression vector p186-GFP. All constructs were transformed into Escherichia coli DH5α cells (TransGen Biotech, Beijing, China), and positive clones were selected and sequenced. The specific primers used in this study are listed in Appendix A.

### 4.5. SlRNC1 Gene Identification, Sequence Alignment, and Protein Structure Modeling

The *SlRNC1* gene was obtained by conducting TBLASTN analysis on NCBI using maize (ZmRNC1), *Arabidopsis thaliana* (AtRNC1), and rice (OsRNC1) RNC1 genes as queries. Subsequently, AlphaFold 2 was utilized to observe the two RNase III domains of SlRNC1. UCSF ChimeraX (version 1.14) was utilized to generate all visual representations of protein structures. Multiple alignment of RNC1 sequences was performed using DNAMAN8 software (https://www.lynnon.com/).

### 4.6. Subcellular Localization Analysis and TEM Analysis

The transient expression vector containing SlRNC1 was introduced into *A. tumefacient* strain GV3101, which was subsequently infiltrated into *N. benthamiana* plants. Two days after agroinfiltration, *N. benthamiana* leaves were examined under a Leica SP8 X laser-scanning confocal microscope (Wetzlar, Germany). To capture GFP images, the fluorescent proteins were excited with the LD laser line at 488 nm. Detection bands were meticulously fine-tuned for each fluorophore group to prevent any unwanted emission overlap, ensuring clear and accurate imaging results.

Transmission electron microscopy (HT7800 RuliTEM, Hitachi High-Tech, Tokyo, Japan) was employed to observe leaves from plants inoculated with TRV:00 and TRV:RNC1 30 days post-inoculation [44]. Briefly, tomato leaves underwent initial processing by cutting them into small fragments measuring 1–2 mm^2^. These fragments were then subjected to fixation by infiltration with a 0.1 M PBS buffer solution containing 2.5% glutaraldehyde. Subsequently, the samples underwent post-fixation in 2% OsO_4_ and were dehydrated using a series of ethanol and acetone solutions. Following dehydration, the samples were embedded in Spurr resin (SPI-Chem Low Viscosity Kit, SPI Supplies, West Chester, PA, USA). The embedded samples were then sectioned using a diamond knife on an ultramicrotome (EM UC7; Leica, Germany), with resulting sections collected on copper grids. Before examination, the sections (1–2 mm^2^) underwent a double-staining process involving uranyl acetate and lead citrate.

### 4.7. RNA-Seq Analysis

Uninoculated systemic leaves collected 24 days after inoculation with TRV:00 and TRV:*SlRNC1* were utilized for constructing RNA-seq libraries and subsequent sequencing. RNA extraction, library preparation, and sequencing were all performed by Novogene (Shanghai, China). Each experimental group, including both the TRV:00 and TRV:*SlRNC1* groups, consisted of three biological replicates, with one plant assigned to each replicate. It is worth noting that samples from the TRV:*SlRNC1* group were exclusively collected from the yellowish regions of systemic leaves. Similarly, samples were collected from the corresponding regions of plants in the TRV:00 group. Differential expression analysis was conducted between the inoculated and control groups (each consisting of three replicates) using DESeq software (v1.20.0), employing criteria of FC > 2 and adjusted *p* < 0.05. Genes exhibiting an estimated absolute log_2_ fold change (log2FC) > 1 or <−1 in sequence counts between libraries, with a false discovery rate (FDR) < 0.05, were deemed significantly differentially expressed [45].

To determine the biological and functional properties of the differentially expressed proteins (DEPs), gene ontology (GO) annotations based on biological process, cellular component, and molecular function were derived from the UniProt-GOA database and InterProScan5 software (https://www.ebi.ac.uk/jdispatcher/pfa/iprscan5). KEGG pathway annotation was carried out using the Kyoto Encyclopedia of Genes and Genomes (KEGG) database.

## 5. Conclusions

The precise role of RNC1 in chloroplast development, as well as the ramifications of chloroplast dysfunction resulting from its malfunction, has not been thoroughly examined. In this research, we successfully identified the ortholog of maize RNC1 in tomato plants, named SlRNC1. Utilizing AlphaFold 2 for protein structure prediction, we confirmed the presence of the two ribonuclease RNase III domains in SlRNC1. Through virus-induced gene silencing (VIGS) of *SlRNC1*, the growth of tomato plants was monitored, and the development of chloroplasts was observed using transmission electron microscopy. Finally, transcriptome analysis was conducted to examine the altered genes and pathways in plants with *SlRNC1* silencing. We discovered that SlRNC1 plays a pivotal role in chloroplast development, likely by regulating the splicing of chloroplast RNAs, but also significantly impacts the growth and development of entire tomato plants by globally regulating gene expression. Specifically, it influences the expression of transcription factors and transcription regulators. Our findings contribute to a better understanding of the crucial role of RNC1 in chloroplast and overall plant development, thereby underscoring its significance in plant physiology.

## Figures and Tables

**Figure 1 ijms-25-06898-f001:**
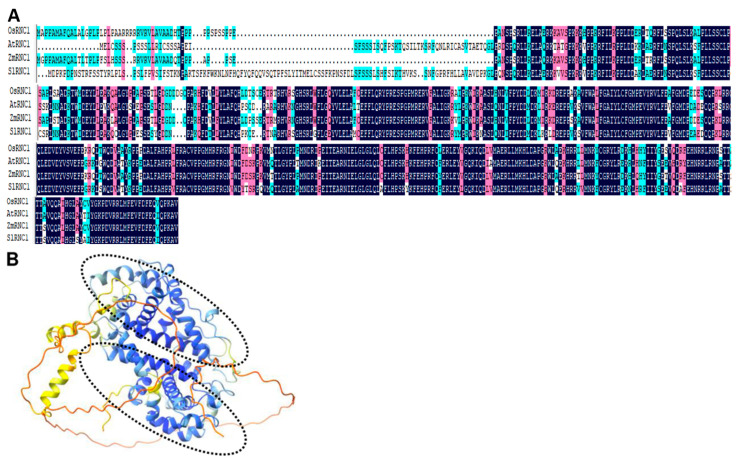
Sequence and structural analysis of RNC1 proteins across four species. (**A**) Multiple sequence alignment of maize RNC1 protein (ZmRNC1) with its orthologs in rice, *Arabidopsis thaliana*, and tomato (OsRNC1, AtRNC1, and SlRNC1). Identical bases are denoted by black shadows, while similar but non-identical bases are highlighted with rose-colored shadows. (**B**) The structural model of SlRNC1 predicted by AlphaFold 2. The two RNase III domains are indicated by two black dashed-line circles. Blue color indicates regions with high confidence, while yellow and red colors indicate low confidence.

**Figure 2 ijms-25-06898-f002:**
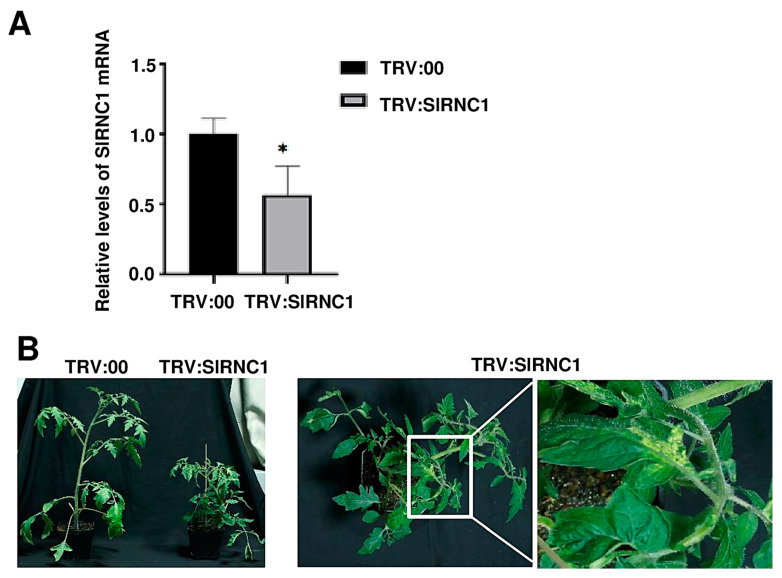
Phenotypes of tomato plants infiltrated with (TRV): 00 and TRV: SlRNC1. (**A**) Confirmation of *SlRNC1* silencing. To evaluate the silencing of the *RNC1* gene, total RNA extracted from systemic leaf samples of the agroinfiltrated plants at 14 dpi was subjected to RT-qPCR. * *p* < 0.1. (**B**) Phenotypes of tomato plants with *SlRNC1* silencing. Two groups of tomato plants were included in the study: the tobacco rattle virus (TRV):00 (mock) group, in which plants were agroinfiltrated with pTRV2 (empty vector) along with pTRV1, and the TRV:*SlRNC1* group, where plants were agroinfiltrated with pTRV2-SlRNC1 along with pTRV1. Phenotypes were recorded at 24 days post-agroinfiltration (dpi).

**Figure 3 ijms-25-06898-f003:**
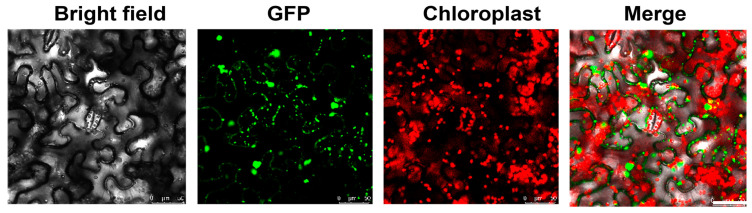
Subcellular localization of SlRNC1. Utilizing the *A. tumefacient* mediated transient expression system, p186-GFP-SlRNC1 was introduced into *N. benthamiana* leaves. Following a two-day incubation period, the localization of SlRNC1 was examined using a Leica confocal fluorescence microscope. The green fluorescence indicate the location of SlRNC1 protein, while the red color indicates chloroplasts. Scale bar = 50 μm.

**Figure 4 ijms-25-06898-f004:**
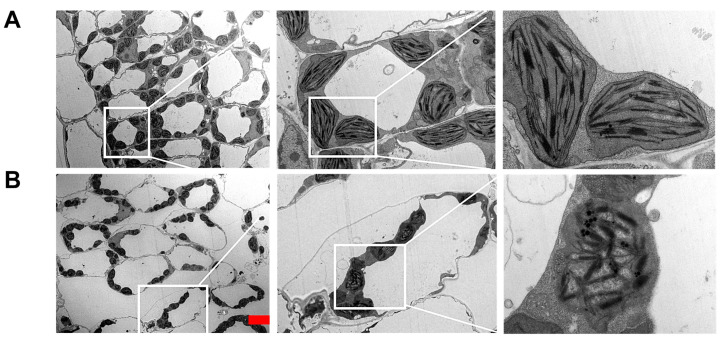
The ultrastructure of chloroplasts from plants in the TRV:00 and TRV:SlRNC1 groups. Transmission electron microscopy (TEM) was used to investigate the ultrastructure of chloroplasts in the leaves of plants inoculated with TRV:00 (**A**) and TRV:SlRNC1 (**B**) at 24 days post-inoculation (dpi). Scale bar = 5 μm.

**Figure 5 ijms-25-06898-f005:**
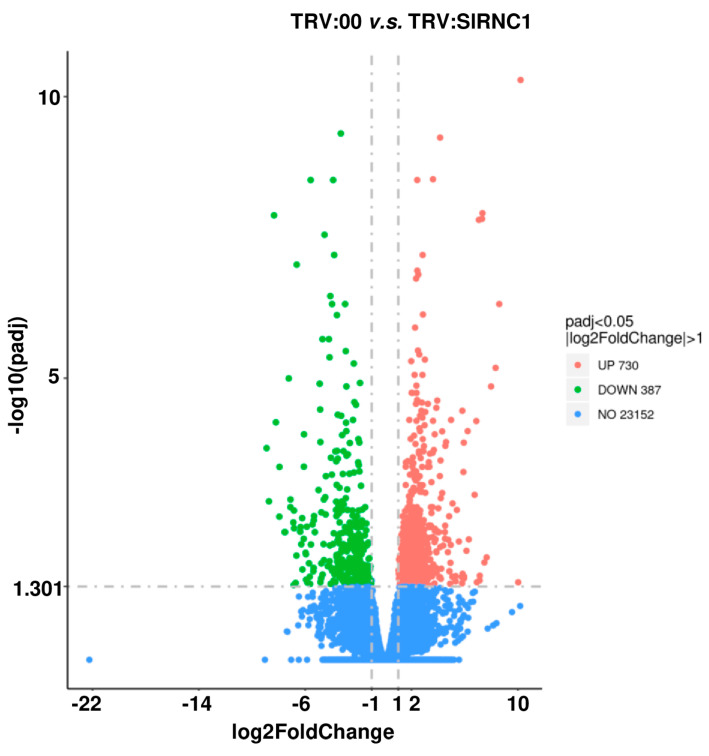
Volcanic diagram of the number of differentially expressed genes (DEGs) between the TRV:00 plants and TRV:*SlRNC1* plants. Each dot represents a differentially expressed gene (DEG). Red dots indicate upregulated genes, while green dots denote downregulated genes.

**Figure 6 ijms-25-06898-f006:**
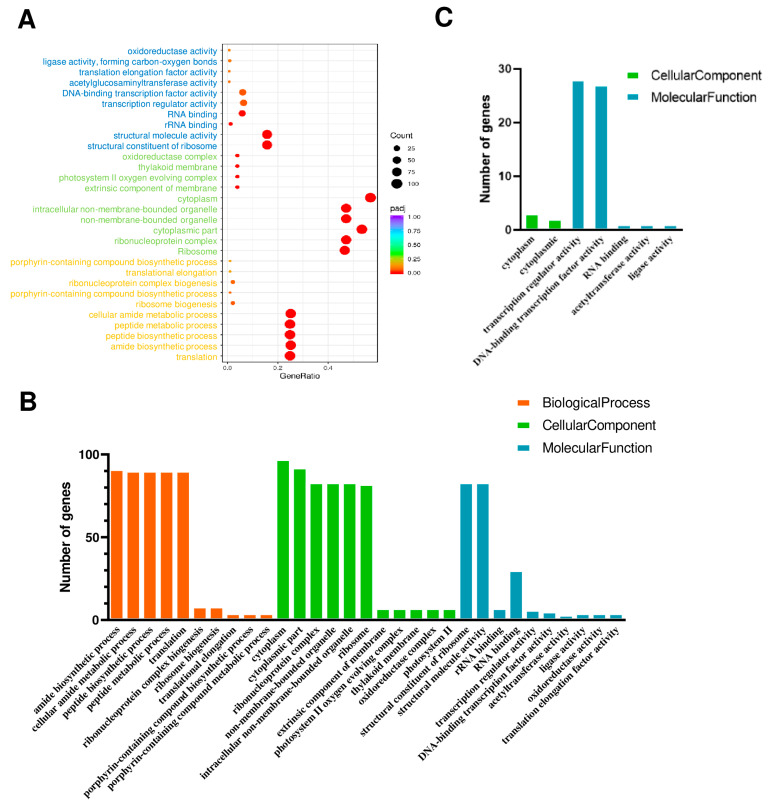
Gene ontology (GO) analyses of DEGs between TRV:00 and TRV:*SlRNC1* tomato plants. (**A**) GO enrichment of DEGs in TRV:*SlRNC1* tomato plants compared to TRV:00. Only the top 10 terms are presented. (**B**) GO enrichment of upregulated DEGs in TRV:*SlRNC1* tomato plants compared to TRV:00. (**C**) GO enrichment of downregulated DEGs in TRV:*SlRNC1* tomato plants compared to TRV:00.

**Figure 7 ijms-25-06898-f007:**
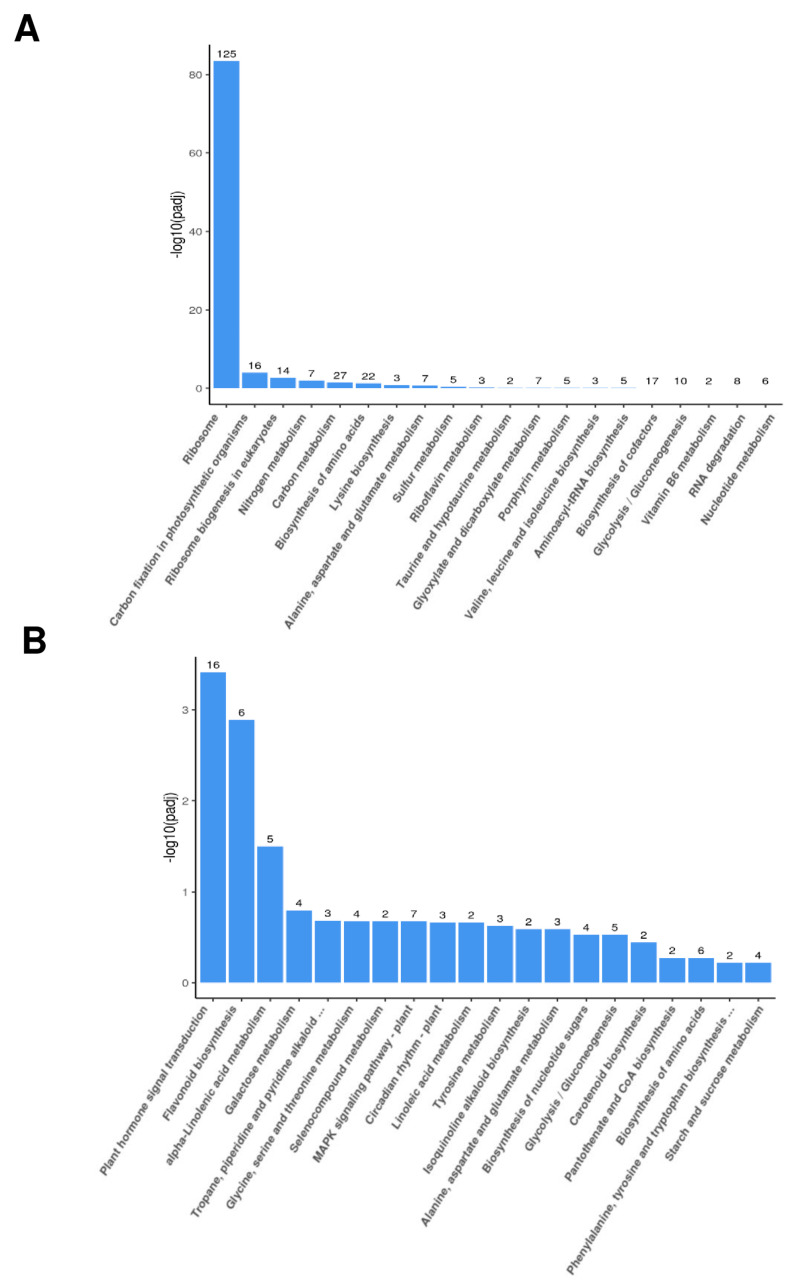
Kyoto Encyclopedia of Genes and Genomes (KEGG) analyses of DEGs between TRV:00 and TRV:*SlRNC1* tomato plants. (**A**) The upregulated genes. (**B**) The downregulated genes. The number above each column indicates the number of DEGs in the enrichment process.

**Figure 8 ijms-25-06898-f008:**
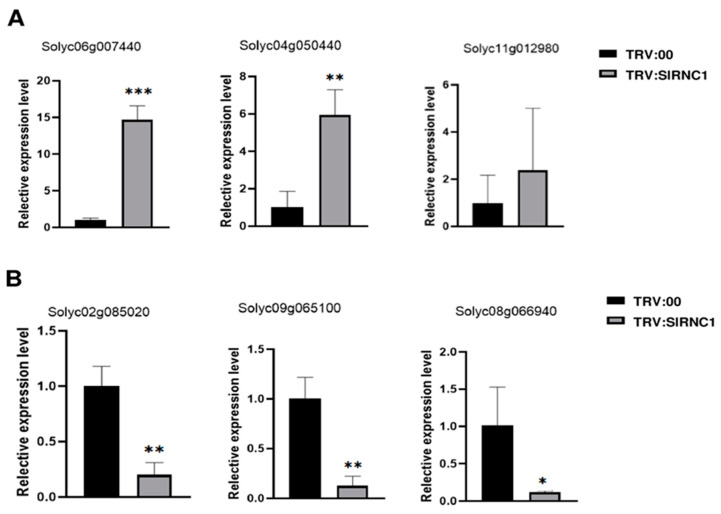
Validation of DEGs by qRT-PCR. To validate the DEGs identified by RNA-seq, three upregulated (**A**) and three downregulated (**B**) genes were further analyzed using qRT-PCR. * *p* < 0.05, ** *p* < 0.01, *** *p* < 0.001.

**Table 1 ijms-25-06898-t001:** Basic information for six RNA-seq libraries.

Samples	Libraries	Raw_Reads	Raw_Bases	Clean_Reads	Clean_Bases	Error_Rate	Q20	Q30	GC_pct
TRV:SlRNC1-1	SRAS240002246-1a	47923554	7.19G	46986592	7.05G	0.01	98.80	96.58	43.48
TRV:SlRNC1-2	SRAS240002247-1a	50772296	7.62G	49650698	7.45G	0.01	98.24	95.70	41.04
TRV:SlRNC1-3	SRAS240002248-1a	46462328	6.97G	43268944	6.49G	0.01	98.83	96.70	43.41
TRV:00-1	SRAS240002249-1a	44465486	6.67G	41516244	6.23G	0.01	98.67	96.26	42.58
TRV:00-2	SRAS240002250-1a	44624792	6.69G	44557430	6.68G	0.01	98.53	95.86	43.27
TRV:00-3	SRAS240002251-1a	45080082	6.76G	43849266	6.58G	0.01	98.80	96.62	42.41

## Data Availability

All data supporting reported results can be found in the published paper.

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
