# Peer review of "Unveiling the Role of SlRNC1 in Chloroplast Development and Global Gene Regulation in Tomato Plants"

_ijms, 2024, doi:10.3390/ijms25136898_

Round 1
Reviewer 1 Report
Comments and Suggestions for Authors
Reviewer 2 Report
Comments and Suggestions for Authors
Dear Authors,
Unveiling the role of SlRNC1 in chloroplast development and 2 global gene regulation in tomato plants
The present manuscript addresses the role of the RNC1 gene in chloroplast development. The experimental design is appropriate, and the studied parameters may interest a broad readership. Many plant scientists are interested in chloroplast development. However, we acknowledge that the quality of the pictures, especially the one from the TEM microscopy, is not optimal. Understanding the importance of clear visual representation, especially regarding critical results, is very important for this manuscript.
The manuscript needs language corrections in several places!
Abstract: Needs language correction. I did correct several mistakes.
Abstract:
RNC1, a plant-specific protein, is known for its involvement in splicing group II introns within maize chloroplast. However, its role in chloroplast development and global gene expression remains poorly understood. This study aimed to investigate the role of RNC1 in chloroplast development and identify the genes that mediate its function in the development of the entire tomato plant. Consistent with findings in maize, RNC1 silencing induced dwarfism and leaf whitening in tomato plants. Subcellular localization analysis revealed that RNC1, including chloroplasts, is localized to the nucleus and cytoplasm. Electron microscopic examination of tomato leaf transverse sections exposed significant disruptions in the spatial arrangement of the thylakoid network upon RNC1 silencing, crucial for efficient light energy capture and conversion into chemical energy. Transcriptome analysis suggested that RNC1 silencing potentially impacts tomato plant development through genes associated with all three categories (biological processes, cellular components, and molecular functions). Our findings not only elucidate the critical role of RNC1 in chloroplast development but also pave the way for further research, inspiring new directions in plant physiology.
Keywords: Are missing here! Please add it here!
Introduction:
Lines 69 to 80 are more suitable for the section Conclusion, which is missing in this manuscript than in the Introduction. Please correct it.
Please formulate the goal of this research and this study here.
Corrected part of the Introduction section!
Plant organelles have a fascinating origin, stemming from two distinct endosymbiotic events [1, 2]. Throughout evolution, these organelles, namely mitochondria and 24 plastids, have engaged in a complex dance of genetic material exchange with the nucleus [3]. This process, known as compartmental DNA transfer, has gradually lost much of their original genomes. As a result of this genetic exchange, chloroplasts have emerged as semi-autonomous entities within plant cells, harboring their own unique genetic material and gene expression systems. Their significance cannot be overstated, as they play a central role in various biochemical processes critical to plant survival. Chloroplasts are veritable powerhouses within the cell, synthesizing essential compounds such as amino acids, pigments, lipids, and plant hormones [4, 5]. Beyond their biochemical prowess, chloroplasts are critical in how plants perceive and respond to their environment. From sensing gravity to regulating stomatal activity and mounting defenses against pathogens [6-9], chloroplasts are integral to a plant's ability to adapt and thrive in its surroundings. They are not just cellular components but dynamic hubs that bridge the gap between a plant and its ever-changing environment. Efficient precursor processing is crucial for maintaining chloroplast translation, as multiple ribonucleases are involved in rRNA processing. Initially, intron-encoded proteins (IEPs) facilitated the RNA splicing of introns. However, in terrestrial plants, nearly all 40 introns have shed their IEPs, leaving only two mature enzymes encoded in the organelle genome alongside a suite of nuclear-encoded splicing factors [10]. Predominantly, the 42 introns found in the organelles of terrestrial plants are of the group II variety [11], which exhibit a catalytic mechanism akin to nuclear spliceosomes. Nevertheless, plant group II introns lose their inherent ability to self-splice and instead necessitate nuclear-coded proteins as cofactors. Since nuclear genes govern most plastid proteins, the nucleus dominates most facets of chloroplast genome expression and Int. J. Mol. Sci. 2023, 24, x FOR PEER REVIEW 2 of metabolism, particularly RNA maturation, including RNA splicing, which heavily relies on nuclear-encoded splicing factors. This communication from the nucleus to the chloroplast is often called a positive signal. Interestingly, the developmental state of chloroplasts can reciprocally influence the expression of nuclear genes. Across various stages of chloroplast growth and development, the nucleus dispatches an array of signals to the chloroplast, thus orchestrating the expression of chloroplast genes and regulating chloroplast development [12]. These signals transmitted from chloroplasts or plastids to the nucleus are called plastid reverse signaling [13, 14].
Material and Methods:
In section 2.1
Please add missing information: the origin of the material used in this study and where the tomato plants come from. Add the botanical name of this plant.
Also, chemical and equipment specifications are missing in this section; please add them.
Several protocols used are missing citations; please add them!
Line 84: The correct botanical name is Nicotiana benthamina Domin. Please correct it.
Line 87: light intensity is missing.
In section 2.2
Please make this title more accurate!
Line 93: Citation for electroporation is missing.
Line 95: Agrobacterium is written with capital A! Please correct it!
In section 2.3
Line 107: Specification for the master kit Vazyme is missing.
In section 2.4
Names of the genes are by nomenclature written in italics and small letters.
Line 119: explain the VIGS abbreviation!
Line 122: Cotyledons and seedlings … Please correct.
Line 123: Citation for inoculation is missing.
Line 128: TransGen Biotech … county is missing.
In section 2.5
Line 138: Arabidopsis, please add the full botanical name of the used plant.
In section 2.6
Line 138: Please write the full name of Agrobacterium. Make these names consistent throughout the entire manuscript! Agrobacterium tumefaciens or A. tumefacient follow the strain name!
Lines 141-142 & 145: Please write the correct specifications for the microscopes, including the country.
Lines 149-155: lack a protocol citation and technical information about the chemicals used, such as Spur resin. Please add all the missing information.
Line 154: what was the size of the sections produced?
Results
Line 172: Specifies all crops by names of the protein sequences used.
Figure 1 is impossible to read!
Figure 2 Out focus: very bad-quality pictures need arrows pointing to the image's critical places!
FiFigure 3 Out focus picture. Very bad-quality pictures need arrows pointing to the image's critical places!
Figure 4 again, lousy quality pictures. The thylakoid structure is not properly visible.
Discussion
Lines 375 and 376: Please add citations for his statement!
Conclusions
Authors forgot this section; please add it to your manuscript!
30.5.2024
Comments on the Quality of English LanguagePlease revise the language in this manuscript.
Reviewer 3 Report
Comments and Suggestions for Authors
This study aimed to investigate the role of Rnc1 in chloroplast development and identify the genes mediate it function in the development of entire tomato plant. RNC1 silencing induced dwarfism and leaf whitening is a good result and could explain the complex shape of twig of tree. I might skip to read; I have interested the localization of this gene in the stress granule (SG).
1) This paper has a value by explaining that RNC1 (Rnc1?) silencing induced dwarfism and leaf whitening. This is an original and good result that could explain why leaves in the shade will die and disappear the complex shape of the twig of the tree or plants appears.
2) I am interested in the localization of this gene in the stress granule (SG). If possible, add the comments in the discussion or show or cite the results.
3) There are small mistakes in the statistics in Tables. The data in the Tables are not consistent: some with 4 digit, some with 3 digit. I recommend it to be consistent with the 3 digit rule.
Overall, this paper is worth publishing in this Journal.

Round 2
Reviewer 2 Report
Comments and Suggestions for Authors
Hello,
It is a pity that the pictures in your manuscript are not of good quality because, for the reader, this is the accurate documentation of your findings and results. Please keep it in mind for the future!
19.6.2023